# Unsupervised Optical Mark Recognition on Answer Sheets for Massive Printed Multiple-Choice Tests

**DOI:** 10.3390/jimaging11090308

**Published:** 2025-09-08

**Authors:** Yahir Hernández-Mier, Marco Aurelio Nuño-Maganda, Said Polanco-Martagón, Guadalupe Acosta-Villarreal, Rubén Posada-Gómez

**Affiliations:** 1Intelligent Systems Department, Polytechnic University of Victoria, Ciudad Victoria 87138, Mexico; yhernandezm@upv.edu.mx (Y.H.-M.); spolancom@upv.edu.mx (S.P.-M.); 2Research Department, Universidad Tecnologica del Mar de Tamaulipas Bicentenario (UTMarT), La Pesca 87678, Mexico; rectoriautmartbis@utmart.edu.mx; 3Division of Postgraduate and Research Studies, Instituto Tecnológico de Orizaba, Tecnológico Nacional de México, Orizaba 92670, Mexico; ruben.pg@crodeorizaba.tecnm.mx

**Keywords:** automatic exam-grading system, computer vision, image processing

## Abstract

The large-scale evaluation of multiple-choice tests is a challenging task from the perspective of image processing. A typical instrument is a multiple-choice question test that employs an answer sheet with circles or squares. Once students have finished the test, the answer sheets are digitized and sent to a processing center for scoring. Operators compute each exam score manually, but this task requires considerable time. While it is true that mature algorithms exist for detecting circles under controlled conditions, they may fail in real-life applications, even when using controlled conditions for image acquisition of the answer sheets. This paper proposes a desktop application for optical mark recognition (OMR) on the scanned multiple-choice question (MCQ) test answer sheets. First, we compiled a set of answer sheet images corresponding to 6029 exams (totaling 564,040 four-option answers) applied in 2024 in Tamaulipas, Mexico. Subsequently, we developed an image-processing module that extracts answers from the answer sheets and an interface for operators to perform analysis by selecting the folder containing the exams and generating results in a tabulated format. We evaluated the image-processing module, achieving a percentage of 96.15% of exams graded without error and 99.95% of 4-option answers classified correctly. We obtained these percentages by comparing the answers generated through our system with those generated by human operators, who took an average of 2 min to produce the answers for a single answer sheet, while the automated version took an average of 1.04 s.

## 1. Introduction

Evaluating the performance of high school students is an issue of concern to government agencies in charge of designing educational policies. Multiple-choice question (MCQ) tests are applicable in assessing different skills and abilities, including higher-order thinking [1]. The education government agencies periodically perform these evaluations to determine the evolution of students’ performance. Government agencies use standardized tests, including MCQ tests, to carry out the assessment. These tests have two elements. The first is a booklet of questions given to the student at the beginning of the test and an answer sheet. The booklet has a fixed number of questions and candidate answers, only one of which is correct, and sometimes the questions are organized in sections, as in the TOEFL test. The answer sheet contains an answer section, where for each question in the booklet, the student must mark only one of the possible answers (usually 4). The answer sheet may have additional elements, such as an identifier to know the student’s identity or other data relevant to the evaluation.

Obtaining answers from an answer sheet image can be accomplished by optical mark recognition (OMR) algorithms based on identifying marks made at specific locations on an answer sheet made with a pen or pencil [2]. The design of the answer sheet can vary from system to system, changing the use of circles or squares to fill the marks and the type of mark considered valid. There are different algorithms, and most use basic image-processing operations, such as thresholding, filtering, contrast enhancement, edge and contour detection, line and circle detection, etc. Patel defines generations of OMR technology. The first includes hardware devices with closed software, which are very expensive and have special requirements regarding paper and answer sheet format. The second generation includes systems that require a computer and a scanner. In the third generation, the OMR migrates to a smartphone using the device’s camera to capture the image of the answer sheet [3]. Even with the recent advances regarding the inclusion of artificial intelligence in smartphones, most currently developed OMR systems belong to the second generation of OMR technology, which favors the control of image-acquisition conditions over the portability of third-generation devices.

We designed the system proposed in this paper for the massive evaluation of exams the educational authority applies to large groups (in this case, the high schools that depend on the Secretary of Education of Tamaulipas). This system avoids the acquisition of expensive hardware and/or software. In addition, the system requires a user interface so that an operator of the educational authority can mass process the answer sheets to obtain the list of answers in CSV format and compare it directly with the output obtained by another system that generates the exams.

The design of the proposed system does not restrict its use only to the educational authority. The teacher of any grade who wishes to use this tool can use it if the answer sheets and exams comply with the requirements established for this project.

The key contributions of this paper are:We proposed a robust algorithm that has been extensively tested in real-world conditions, where there is significant variability in the applicants’ answer styles. The algorithm we implement does not require a training stage for exams with the same layout; hence, it would work without any changes, as demonstrated for the dataset with more than 6000 real-world answer sheet images, correctly classifying more than 99% of the test items in the dataset.We developed an OMR system based on well-known image-processing algorithms, which enables us to understand what happens at each stage of processing, locate errors if they occur and identify their possible causes. The latter is not possible with the use of more complex algorithms, such as those based on convolutional neural networks. We developed a system that does not rely on complex algorithms for training and does not require high-performance hardware, making it suitable for computers with modest computing power, such as those found in office environments.The average execution time (1 s per answer sheet) is fast enough to be included in an automated exam-evaluation chain. Although it is not a real-time algorithm, we could reduce its execution time using process-optimization tools and high-performance hardware.

## 2. State of the Art

We reviewed the state of the art to identify the most relevant works. Many present advances but only make qualitative measurements and not quantitative ones. Many works do not establish dataset conditions or demonstrate variability with respect to illumination conditions. Many focus on creating a generic system for multiple test formats, which would be useful for any teacher regardless of the number of tests they will perform. Others have focused on migrating such a system to web platforms or smartphone applications. Regarding mobile apps, many authors explored the complexities of an open environment where any picture taken with a smartphone without lighting and orientation restrictions could lead to greater complexity in detecting the marks.

### 2.1. Controlled Condition Answer Sheet Acquisition

Some authors propose a scheme similar to the one proposed in this article. They use a personal computer with a scanner to obtain the digitization of the answer sheet, which is beneficial since the system acquires images under controlled conditions. Many works report high accuracy, with few answer sheet images or even no mention of the accuracy and dataset features. In [4], the authors proposed a Matlab script running on a PC with a scanner to evaluate MCQ tests such as those used in exams like TOEFL. The exam format used to evaluate the system has 15 questions with four choices for each question, and it tested to be robust with the type of pencil used to answer the test. In [5], the authors develop an image-processing-based system to automatically obtain the responses from MCQ printed answer sheets, including the sending student feedback e-mail as an additional feature. They measure the minimum time to process the marks, obtaining 1.4 s to process an answer with an accuracy of 100%. In [6], the authors proposed an approach for automatic paper grading, which deals with variations of marks made by the students in the answer sheets.

Few systems report a massive use for obtaining the answer sheet responses. In [7], the authors developed software for process and evaluating MCQ answer sheets. They tested the system using 4000 exams, achieving reliability with different answer sheets and filling out patterns in the processed answer sheets. In [8], the authors developed a PC-based system with a portable scanner to correct handwritten answer sheets automatically. They used 250 images captured from tests done by Prince Mohammad Bin Fahd University students and trained two CNNs to recognize handwritten characters. They evaluated their tool’s performance through the accuracy metric, yielding a maximum of 92.86% when comparing predicted vs. real answers. In [9], the authors proposed a system to score specific question types automatically in paper-based tests using artificial intelligence techniques. To evaluate the system, they created a 734,825 multiple-choice answers dataset, tested for show answer type and obtained a recognition accuracy of 95.40%. In [10], the authors report the need to apply an automatic evaluation of 700 paper tests related to programming courses in the Electrical Engineering and Computing study program at the University of Belgrade, School of Electrical Engineering and propose a Python-based tool.

### 2.2. Uncontrolled Condition Answer Sheet Acquisition

Other work focuses on implementing an open system, where acquiring the answer sheet image is possible using smartphone or web cameras. The latter complicates the system, as the lighting and posing conditions vary, requiring operations that require more computing power. In [11], the authors developed a mobile app that uses image-processing techniques to extract answers from MCQ test images acquired through the smartphone’s camera. In [12], the authors report an image processing-based that captures answer sheets from a web camera, applies basic image-processing operations (such as Canny edge detection) to locate the answer responses and generates a grade based on the student’s responses. They evaluated the system performance, obtaining an accuracy metric of 94%. In [13], the authors proposed a web application for correcting and grading MCQ tests that use OMR to extract the answers from the printed answer sheets. The app adds an encryption scheme to protect the digital information of assessed students through QR codes, and the system processes the scanned answer sheets. In [14], the authors developed an image processing that deals with issues acquiring answer sheet images, such as wrong orientation and blur of irregular illumination. They also replace scanners with smartphone cameras to reduce costs. In [15], the authors proposed a webcam-based OMR using simple preprocessing, segmentation and recognition algorithms, and tested the system with variations in orientation, light conditions and layout of the answer sheet. In [16], the authors proposed using a webcam to extract responses from an OMR print answer sheet. They evaluated several formats of MCQ and implemented an image-based matching algorithm to extract the responses. They evaluated the proposed system on three formats of MCQ tests, obtaining an accuracy of 97.6%.

### 2.3. Classical Computer Vision-Based Systems

Most papers attempt to solve the problem of answer sheet recognition in MCQ tests solely by computer vision techniques without requiring machine learning algorithms. In [17], the authors propose a correction method for MCQ based on mathematical morphology, thresholding and neighbor operations. The authors claim that their system improves the recognition rate even for incompletely marked options, impacting the overall correction accuracy. In [18], the authors developed a system to automatically grade MCQ by developing an OMR algorithm based on digital image-processing operations such as color conversion, thresholding, morphology, connected components analysis, etc. In [19], the authors propose a mark-detection algorithm on bubble and box fields and claim to improve existing thresholding and simple counting-based algorithms, testing under several variations, including print and scan artifacts. The authors evaluated the proposed solution using sensitivity, specificity and accuracy metrics. In [20], the authors developed an optical mark recognition (OMR) to automate the evaluation of simple and complex questionnaires used in university admission. They recognize optical marks by tuning sheets and mark sizes, allowing them to locate margins to detect mark localizations and estimate groups or columns. They tested different configurations to validate several column variations using clear digitalized images. In [21], the authors proposed an OMR algorithm designed to process smartphone-captured images that address problems such as illumination variation. They compared the proposed algorithm versus the existing Pixel Projection-based and Hough transform-based methods and tested them using shadows and illumination variations. In [22], the authors developed an algorithm for detecting shape defects caused by color and resolution to avoid false detection in filled shapes in MCQ answer sheets using an arithmetic sequence that estimates the undetected circle-shaped locations. In [23], the authors developed a system for detecting OMR marks. They used standard image-processing techniques such as Gaussian filtering, OTSU thresholding, morphological operations and contour-detection operations. Finally, they compared their system performance using the F1 and accuracy, reporting a maximum of 93.33% for the last. Some previously mentioned articles that use computer vision operations [4,5,6,11,12,13,14,15,16].

Other authors have attempted to solve the recognition problem using an unsupervised approach. In [24], the authors introduce an unsupervised framework named the Single Mark Localization Model (SMLM), which addresses problems in optical mark recognition. Their framework used an energy optimization-based algorithm to locate optical mark localization, which processes the pixel distribution, regional pixel proportion and aspect ratio. They evaluated the proposed framework for the precision and recall metrics, obtaining 99.82% and 99.03%, respectively, for a dataset of 18,000 answer sheets containing 82 types of marking symbols.

### 2.4. Deep Learning Approaches

Recently, recognition tasks have used deep learning algorithms. There is research work where the answer sheet response recognition uses DL-based techniques. In [25], the authors use the YOLOv8 object-detection model to detect and classify OMR marks. They combined the YOLOv8 object detection with a DBSCAN clustering algorithm to group columns and automatically estimate the question order. The system deals with unanswered and multiple-marked questions. They trained and evaluated the model using the precision, recall and mAP (mean average precision) metrics, obtaining 96.5%, 99.8% and 97.3%, respectively. In [26], the authors proposed an image-registration algorithm to obtain answers from MCQ answer sheets by training a classifier to recognize three types of answer marks. They compared both feature extraction and CNN-based approaches. In [27], the authors propose a transfer learning-based CNN model, which classifies answer boxes into confirmed, crossed-out and empty classes. The proposed CNN uses a MobileNetV2 architecture with four fully connected layers. The authors used a five-fold cross-validation and obtained an accuracy of 95.96%. In [28], the authors used a fast object-detection model to locate markers and divide the answer sheet into smaller regions using PC. Each area is processed later to estimate the user’s response. Their system can process answer sheets in 20 ms, obtaining an error of at most 0.005%. Other papers also used DL-based approaches [8,9].

## 3. Materials and Methods

### 3.1. Answer Sheet Folder Organization and Layout

The operator must organize the images of the scanned answer sheets into three levels of folders. The root folder contains one or more folders, each corresponding to a school or institute where the test was applied, and the application assumes the folder’s name as the institution’s name. Each institution should have at least one folder for each grade, generally 3, since Mexico has a 3-year high school format, and the application is applied once a year. Each grade folder must have at least one image in standard image format (png, jpeg or tiff). Figure 1 shows the directory structure described above.

The state education authority of Tamaulipas requires a system to extract the answers massively from answer sheets captured using standard scanner devices. Figure 2a shows an example answer sheet, which has the following layout:A space where the students handwrite their name and date of exam application.A first reference rectangle with four rows of circles ranging from 0 to 9, which will serve to establish a 4-digit identifier.A second reference rectangle containing three columns with items having four options for each question, grouped depending on the grade to which the test is applied. In one case, we have two tests: one with 90 items and one with 100. In the 90-item test, we found three columns of 30 items each, while in the 100-item test, we found two columns of 33 and one column of 34 items. Figure 2b shows the localization of these reference rectangles in a sample answer sheet. The second reference rectangle has three circle option columns, named section 0, section 1 and section 2.

The response-detection process might seem trivial; however, the current version faced different issues described below:A standard error is that the sheets become bent during scanning.Different scanner conditions were detected, such as resolution and image quality variations, which, if not corrected, would generate errors when recognizing the responses.The exams may not be oriented correctly, so preprocessing must be added to correct incorrect orientation scans.Even when the exams are correctly oriented, minor rotations could affect recognition performance, so an angle correction process is required to correct the last.

The following describes the dataset obtained from Tamaulipas’s Subsecretaría de Educación Media Superior y Superior, which is the authority that evaluates through multiple choice exams high school students, which correspond to the second, fourth and sixth grades, equivalent to 10th, 11th and 12th grades in the US. The total number of test images is 6029, which come from 44 campuses of all the high schools in the state. Table 1 summarizes the previously described dataset.

The education authority schedules the exams for specific dates. The tests, along with blank answer sheets, are sent to the schools that apply the test and scan the answer sheets for their return to the education authority. Each school applies and is responsible for digitizing the exams, and this digitization does not follow a specific standard. We note that the digitization of the exam by multiple sources can lead to different difficulties when processing the images of the answer sheets.

### 3.2. Proposed Algorithm

Figure 3 shows the main dataflow of the proposed system. The algorithm begins by opening a root directory organized in directories named according to the defined in Section 3.1. The algorithm walks through the directories, opening each test file inside them. The algorithm then applies the following operations to each image in the grade directory. We first open the image in BGR space. Next, we force the orientation of the sheet to portrait, by rotating the image 90° anticlockwise if the column dimension of the image is greater than the row dimension. In the next step, we resize the image to a fixed size. In the particular case of this work, we defined a fixed dimension of 1670×1290 pixels. The latter allowed us to standardize the values of several thresholds needed by the proposed algorithm, preserving the ratio of the letter paper size (1.294:1) used to print the physical test sheets. In the following operation, we converted the BGR image to grayscale, taking care of storing a copy of the BGR image to present results at the end.

Next, we detect the contours of the rectangles surrounding the ID and answer sections through the ID and Answer Section-Detection Process (IDASP). It performs image-preprocessing operations to reduce noise and correct illumination, then thresholds the image to detect rectangles containing the ID and answer marks. Depending on the detected reference rectangles, we verify the exam sheet orientation by analyzing the order of the answer and ID sections. If the exam sheet is inverted, we flip the sheet by 180° and execute the IDSAP once again. We then match the coordinates of the detected rectangles’ corners with those of the reference image’s rectangles and compute a homography to apply a perspective transform to the entire image, correcting for any rotated images. After this perspective correction, we re-detect the reference rectangles using IDSAP to obtain the final reference rectangles used in the remainder of the algorithm.

At this point in the algorithm, we have the two reference rectangles, where the first rectangle (index 0) corresponds to the ID section and the second rectangle (index 1) corresponds to the answers section. We first process the answer section. After we detected the reference rectangles, we obtained the thresholded versions of the S and V channels of the HSV version of the initial BGR image, as we use in several further processes. We process the area inside the ID and answers sections rectangles differently because the size of the option circles and the separation between rows and columns of the circle matrix are different. The first step in this process is the definition, by distance to the reference rectangle, of three bounding boxes corresponding to three columns of answer circles. We then process each bounding box separately by creating a background mask to select (by a bitwise and operation) the corresponding answer options in a binary image obtained by an adaptive thresholding of the normalized image obtained after uneven illumination correction. We use a thresholded version of the H channel of the HSV image to evaluate if the zone around the answer circles is too noisy. We use the bounding box mask to select the region of interest and obtain the difference between the thresholded image of the *H* channel and the thresholded answers. The resulting image would contain the pixels corresponding only to the background of the current bounding box. When the number of pixels in the background of the bounding box in the binary image is greater than 15% (selected experimentally) of the area of the bounding box, we consider it too noisy and apply the cleaning process described in process AIDSCMD.

Once the image has been analyzed to detect its orientation and resized to the predefined size, it is necessary to detect the ID and response sections using two sequential processes. It is required to assign an index to the ID and answer regions, apply HSV conversions and normalize the S and V channels. Once finished, the Answer Section Processing (ASP) and ID Section Processing (IDSP) processes analyze the ID and response regions. When both processes finish, a final block generates the XLSX and TXT files. Figure 4 and Figure 5 illustrate the dataflow for both ASP and IDSP processes.

The ASP process defines the bounding boxes within each section, creates a mask, counts the pixels, compares them concerning a threshold and removes noise if needed. It then detects the contours, groups them and generates groups of four circles to create the option lines. This process executes NBSP, AASCMD, COPCRI, CCIMBGI and AVTLD processes. Figure 4 shows the dataflow within this process.

The IDSP process performs various morphological operations, counts the pixels of the obtained regions and compares them with a preset threshold. It also detects the contours of circles and can perform denoising processing. Additionally, it sorts the circles, groups them into 10 elements from each row of the ID section and assigns them to corresponding IDs. Based on the latter, it determines whether the assigned answer is a number corresponding to the marked circle for the current row, a multiple selection (M) if there is more than one mark or no selection (X) if there is no mark. This process calls to NBSP, AIDSCMD, COPCRI and CCIMBGI processes. Figure 5 depicts a dataflow for this process.

The next stage in the algorithm is to count the number of circles inside the current bounding box, but we need to consider some aspects. We can find two types of tests in the used dataset: 90 or 100 items. In the 90-item tests, we found three columns of 30 questions each, with four options, presented as letters surrounded by circles. In this case, we will have 120 circles per bounding box. On the other hand, in the case of the 100-item tests, the first and second columns have 33 questions, while the third column has 34, with four options per question, giving us 132 circles for the first and second columns and 136 for the third one. To verify if the number of detected circles is correct, we verify if the number of circles in the current bounding box equals 120, 132 or 136. If the number of circles differs, we apply one of the alternative processes defined in process AASCMD. Once we have the required number of detected circles, we order them in each bounding box and prepare the reference number of pixels and mean grayscale values we will use to evaluate the answers. The detailed operations are described in process COPCRI. Next, we gather the ordered circles in groups of 4, corresponding to the rows of the matrix of circle options inside each bounding box. In the next step, we compare the evaluation results of the detected marks in each row of the circle options in the current bounding box. We perform this evaluation following the process described in CCIMBGI process. After we obtain the marked circles in each row of the circle options, we define the value corresponding to the selected answer by the analysis described in the process NBSP. This process also receives information about the ID number, which we store in a file once we define the answers layout. After storing results in files, we finish the program execution.

The processing of the ID section (index 1 of the main rectangles) begins by applying morphological closing and opening operations with a small structuring element to eliminate holes and small pixel clusters. Next, we verify if the number of pixels in the background inside the bounding box in the binary image is greater than 15% of the area of the bounding box. When the image is too noisy, we apply the cleaning process described in process AIDSCMD. The next step is the detection of the option circle contours. We then pass the detected contours to the process NBSP, where we count these circles and verify if we have the correct number of them. The ID section has four rows of ten options, summing to 40 option circles. Process AIDSCMD yields an array of 40 circles.

Once we detect the answer circles, we order the circles from top to bottom and from left to right, following the process described in process COPCRI. This process also prepares the reference values from the binary and grayscale values we use to compare the obtained values while evaluating the selected options in each row of the circle answers. Using the ordered circles, we gather them in groups of ten and compare the identified circle marks with the binary and grayscale reference values, as we describe in the CCIMBGI process. We count the number of detected marked circles in the ten options of each row. If we detect more than one marked circle, we assign an M (Multiple) value as the selected answer. If we do not detect a marked circle, we assign a value of X. If we only detect one answer, we assign the selected answer a value between 0 and 9 based on its index position. Finally, we concatenate the answers obtained for the four rows and pass the information to the process that writes the results in the .txt and .xls files.

#### 3.2.1. ID and Answer Section Detection

Figure 6 illustrates the dataflow for detecting the ID and answer sections IDASD. This process detects the contours of the two largest rectangles in the test image by applying an image blur with a 3×3 median filter to slightly remove noise, without creating new intensity values. The following step is the correction of uneven illumination by pixel-wise dividing the original grayscale image by approximating the illumination gradient fabricated using a morphological grayscale opening operator with a 25×25 rectangular structuring element. Next, we obtain a binary image by thresholding using Otsu’s algorithm. We then use a morphological closing with a small 3×3 structuring element to avoid holes on the border of the external rectangles to detect. The final step in detecting main rectangles is applying a contour-finding algorithm with a minimal set of contour points. If the number of detected contour points is 4, we assume the rectangle is successfully detected; if not, something went wrong. We tested the number of found rectangles (4-point contour sets) equal to 2. The latter process works well for an ideal situation where the test was cleanly answered, without putting text out of the designed zones or having dark spots produced by a deficient erasing technique. It is fast and requires few operations. However, there are several tests where the assessed student writes over the border of the external rectangles or inside them, affecting the border-detection process. We defined the three alternative processing techniques described next to cope with these situations. If the number of found rectangles does not equal 2, one of the following processes is used to test in the numbered order in which they appear.

Alternative rectangle finding process 1. If the main process fails to detect two rectangles, this is the first alternative process to test. This process implies more operations and uses more copies of the images to work, but it allows us to detect the two main rectangles when the assessed student wrote over the border of the rectangles. In this alternative process, we initially detect the contours over a thresholded version of the normalized image obtained after the uneven illumination correction. In this case, a low intensity value is set as a threshold (60) to capture the grayscale values corresponding to the borders. Next, we detect the contours of the resulting binary image and fill the regions inside the contours with the maximum intensity value. We apply an iterative morphological opening operation with a medium-sized structuring element (9×9 in our case) to the resulting filled polygons to eliminate noise and smooth the borders of the binary objects. Through experimentation, we found that the minimum number of iterations that yielded clean binary objects was 7. The next step is a new detection of the binary objects’ contours, filtering by area and keeping only the contours with an area greater than 2% of the area of the image. Due to the aggressiveness of the applied morphological processing, these contours might contain holes. To cope with these discontinuities, we approximate the contour polygons using the Ramer–Douglas–Peucker algorithm [29].Alternative rectangle finding process 2. We found that the first alternative rectangle-finding process could fail if the assessed students wrote text inside the ID or answer bounding boxes. For those cases, we defined the following steps. First, we thresholded the image using the Otsu algorithm to select the threshold. As the next step, we apply a morphological opening with a small structuring element (5×5) to eliminate small pixel clusters. The following steps are similar to those used in the first alternative process; however, before detecting the external contours, we apply an iterative morphological opening with a medium-sized structuring element (9×9). In this alternative process, the number of iterations is 5. The last two steps are the same as applied in the first alternative process.Alternative rectangle finding process 3. This alternative process allowed us to correctly process images where some sections of the contour of the two main rectangles were missing (significant discontinuities). In this version, the first three steps, thresholding, morphological opening and external contours filtered by area, are the same as those used in the second alternative process. The next step is the detection of the extremes of the detected contours, by ordering the contour coordinates and finding the lowest and largest coordinate x and y values to define the corners of the rectangles. We also verify that the distance between corresponding corners of each side of the rectangles is within a range previously measured over a typical sample. We then convert the corners into the required contours. This process is computationally costly compared to the main process.

If, after the defined alternative rectangle finding processes, we do not have the expected two rectangles, the process throws an error and finishes the execution of the program.

#### 3.2.2. Alternative ID Section Circle Mark Detection (AIDSCMD)

This process receives the detected circles in the ID section. If the number of circles equals 40, this process ends. Otherwise, we apply one of the following alternative processes.
Alternative ID section processing 1. This process works well when the assessed students chooses the option with a big mark touching the border of the external rectangle of the section. In this process, we work with the initial binary image obtained in the main process masked with the current bounding box. First, we apply an iterative morphological opening with a 5×5 structuring element to ensure a separation between the circles and the border of the bounding rectangle of the section. Next, we detect the external circle contours, filter them by area and fit a minimum enclosing circle. Finally, we count the number of resulting circles.Alternative ID section processing 2. This process allowed us to work with images where the assessed student filled the option circles with a mark that was too big, extending beyond the circle border. In this alternative process, we also work with the initial binary image obtained in the first stage of the main process masked by the current bounding box. First, we dilate this image with a horizontally oriented 15×3 rectangular structuring element. Then, we produce a second version of the binary image by dilating the initial binary image with a vertically oriented 3×15 rectangular structuring element. Next, we perform a bitwise and operation between the binary image’s vertical and horizontal oriented versions to obtain the crossings. To eliminate holes, we apply a morphological closing with a 3×3 structuring element to the resulting image. Next, we detect the contours of the resulting binary image, filter by area and fit minimal enclosing circles to the contours. Finally, we count the number of resulting adjusted circles.Alternative ID section processing 3. This process allowed us to correctly process images where the assessed student used a deficient erasing technique, producing stained circle marks. We begin by applying an adaptive thresholding to a median-filtered version of the grayscale image. Next, we apply a morphological closing using a 3×3 structuring element. Then, we detect the contours, filter by area and fit minimal enclosing circles. Finally, we count the number of resulting circles.Alternative ID section processing 4. This process worked well with images where the assessed student marked the option circles using different intensities and marks extending from the pretended circle area. In this case, we work with the saturation channel, *S*, of the HSV version of the input image, masked by the current bounding box. Next, we apply an iterative morphological closing with a 5×5 structuring element and then detect the external contours. We use the detected contours to draw them filled over a new binary image. We apply an iterative morphological closing over this image using a 3×3 kernel, extract its external contours, filter them by area and fit minimal enclosing circles. Finally, we count the number of adjusted circles.

If the number of circles differs from 40 after the four alternative processes, we throw an error and the program stops. Figure 7a depicts a dataflow diagram of the process described above.

#### 3.2.3. Alternative Answer Section Circle Mark Detection (AASCMD)

In this process, we first verify if the number of circles obtained from the main process in the current bounding box is correct. If not, we apply one of the alternative processes we next describe. Figure 7b depicts a dataflow diagram of the process described below.

Alternative answers section processing 1. This process works on images with variations in the grayscale levels affecting the thresholding process used in the main process. The process begins by thresholding a slightly blurred version of the RGB image around the color (in the RGB space) corresponding to the circles of the answer options (orange in our case). The following operations are similar to the second part of the main process, where the objective is to obtain the crossing between two versions of the binary images. To obtain the first version of these images, we applied a morphological closing operator to the answer circles using an enlarged structuring element in the vertical direction. Similarly, we apply the same morphological operator to the answer circles using an enlarged structuring element in the horizontal direction to obtain the second version. We then use morphological opening and closing operations to eliminate holes and spur pixels around the detected circles. Next, we compute the centroids of the detected crossings and draw filled circles with a radius equal to the radius of a reference circle on a clean sheet. Finally, we count the obtained circles.Alternative answers section processing 2. We found that for some images, when the assessed student marks the answer circle with a too big mark invading the space of the surrounding options, the main and first alternative processes might fail. If this happens, we apply a different method that uses an adaptive thresholding over a median filtered version of the original grayscale image. Next, we obtained the region of interest by masking with the current bounding box. After a morphological closing with a 3×3 rectangular structuring element, we obtain the external contours and draw them filled. We correct holes and spur pixels with morphological closing and opening operations before detecting external contours again. We filter the detected contours by area and use them to fit a minimum enclosing circle around them. The fitted contours represent the answer circle options, and we count them to estimate the number of answers in the bounding box rectangle.Alternative answers section processing 3. This process allows us to process images where the assessed student marked the answers using different circle sizes and grayscale tones. In this case, we used the input image’s saturation layer (S) of the HSV image. As the first step, we threshold the *S* layer using Otsu’s algorithm and apply the current bounding box mask. Then, we apply an iterative morphological closing using a 5×5 square structuring element. Next, we find the external contours and draw them filled out. We correct the borders of the filled contours using a morphological closing using a 3×3 structuring element and detect the contours again. We filter the detected contours by area and fit a minimum enclosing circle to the detected contours. Finally, we count the fitted circles.Alternative answers section processing 4. We found that in some tests if the assessed student made a mistake, he could eventually erase his mistake, but also the circle mark. The latter could cause the main and previous alternative processes to fail. This alternative process begins thresholding the normalized image obtained after the uneven illumination correction using a threshold close to the maximum value to ensure capturing the borders of the available circle options. After masking the current bounding box, we detect the contours inside the bounding box, filter by area and detect the centroids of the contours. We detect the nearest centroids to the corners of the bounding box and compute the vertical distance to evaluate if the number of answer rows in the current section is 30, 33 or 34. We then create a matrix of circles with 30, 33 or 34 rows and four columns inside the current bounding box and with the same radius as a reference circle detected in a clean test sheet.

If, after the alternative processing described above, the number of circles in the current bounding box differs from one of the three established values, the process throws an error and the program stops.

#### 3.2.4. Circle Ordering and Pixel Counting in Reference Image (COPCRI)

We first detect the circles in the left and right corners of the top row to order the detected circles from top to bottom and from left to right. We then compute the distance of the center of every circle to the line traced between the centers of the circles in the top corners of the matrix, and we use this distance to order them by the y coordinate. At the end of this operation, we order the circles from left to right, depending on their x coordinate. At the end of the ordering process, we extract the reference values of the number of pixels inside each option (A, B, C or D) in a binary image of a clean test sheet. We also compute the mean grayscale value in the V channel of the HSV image of the input image in the areas inside each answer option. These values would be used as reference values to compare the assessed student’s marked answer. Figure 8a depicts a dataflow diagram of the process described above.

#### 3.2.5. Cross Comparison of Identified Marks in Binary and Grayscale Images (CCIMBGI)

We evaluate the marked answers following two possible paths. First, we threshold the normalized image obtained after uneven illumination correction with a high threshold to capture the grayscale levels corresponding to the option circles. Then, we extract the pixel intensities inside the area of the detected circles for each row of the option circles in the current bounding box of the thresholded image and the V channel of the HSV image. We sum the pixel values extracted from the binary image, compute the mean of the grayscale values for each option circle and store their values in a 4-length array, one for the binary values and one for the grayscale values. In the case of the binary values, we assign true to the corresponding array index if the number of pixels in the detected answer exceeds the reference value. Otherwise, we assign false. In the case of the grayscale values, we assign true to the corresponding array index if the computed mean of grayscale values is lower than the reference grayscale value, meaning that it is darker than the reference circle. The information of the two boolean arrays is combined using an and bitwise operation. Figure 8b depicts a dataflow diagram of the process described above.

#### 3.2.6. Noisy Background Section Processing (NBSP)

In this process, we isolate the answer circles by dilating the circles inside the bounding box mask using structuring elements of 15×3 and 3×15 in the vertical and horizontal directions, respectively. We also apply a morphological closing with a small structuring element to eliminate holes. Next, we use a bitwise operation between the vertical and horizontal versions of the images. The resulting image contains the crossings corresponding to the rough location of the answer circles. Next, we filter the resulting binary image by area to eliminate small pixel clusters. Then, we compute the centroids of the detected crossings and use this information as the center of a circle with a radius equal to the length of the radius of a reference circle in a clean sheet. We use the centroid and radius of the circles to draw filled circles, ignoring the pixels in the background. The final step in this process is counting the resulting circles. Figure 9a depicts a dataflow diagram of the process described above.

While it is true that we experimentally estimate the intensity and size thresholds of the morphological kernels, these thresholds depend on the dimensions of the bounding boxes and the average vertical and horizontal distances between the circles of the options (items) in the answer sheet format.

To determine the threshold selection that determines the presence or absence of noise within the bounding boxes, we analyzed the number of pixels in the background of a set of scanned sheets without response marks using near-ideal illumination conditions. We averaged the number of pixels for these clean images.

#### 3.2.7. Answer Values by Row and Test Layout Definition (AVRTLD)

In this process, we assign readable values to each row of option answers according to the evaluation results of the rows of four circles. First, we check if the number of detected marked circles exceeds one. If true, we assign M (Multiple answer) as an answer status value. Next, we verify that the number of selected answers equals 0. We assign X (unanswered) as an answer status value in this case. If none of the previous conditions occur, we assign A, B, C or D according to the index where we found the answer mark. The last step in this process is to count the number of rows in the current bounding box. If the number of rows equals 30 in each bounding box, we conclude that this test contains 90 questions. If the two first bounding boxes contain 33 rows, we conclude that this test contains 100 questions. We use this information to define the layout when storing the results in the txt and xls files. This process also receives the information about the ID number found in another branch of the algorithm that is included in the files to write to disk. Figure 9b depicts a dataflow diagram of the process described above.

### 3.3. Automatic MCQ Answer Sheet Recognition Application

To implement the proposed application, we employ the following software development tools:Python. This language allows us to develop multi-platform applications with minimal effort and provide object-oriented programming. The version used is 3.10.12.OpenCV. This library allowed us to implement the image-processing algorithms needed to identify the responses of the digitalized answer sheets. The version used is 4.11.0.PyQt6. This library allowed us to add graphical user interfaces (GUIs), including functions to select a folder graphically. The version used is 6.7.0.

Figure 10 shows the proposed software application for mass image processing of multiple-choice question paper exams. The application has three panels: left, center and right. The top of the left panel has a button allowing the user to select the root folder or up to two levels down in the folder’s hierarchy containing exams, as defined in Figure 1. There are three options:The user selects the root folder with images associated with each institution where SET applied the exam, each institution folder having at least one subfolder.The user selects an institution’s folder inside the root folder. The app requires a process to identify the parent folder but only includes the designated institution’s information in the interface.The user selects a grade folder inside one institution; being necessary that the interface infers the institution of the parent folder and the name of the root folder from the parent of the institution folder.

The left panel includes several controls in addition to the folder open button mentioned above. Once the user has selected a folder, two combo boxes display information about the chosen folder, one with the institutions and the other with the grades detected within those institutions. A listview appears at the bottom of the left panel, which contains a list of the answer sheet image names in the selected directory. The user can change institutions by choosing a different option from the institution combobox, and the interface updates the grades combobox and the images listview. The right panel contains a preview of the current answer sheet image, which can be zoomed using mouse events (by default, the first answer sheet images of each folder, sorted in alphabetically ascending order). The interface updates the preview image if the user changes the listview selection.

In the central part of the application, there is a panel that displays information on the total number of answer sheet images and the institutions contained in the root folder. Another panel displays detailed information on the current institution, including the detected grades and the total number of answer sheet images for that institution. The central panel includes two buttons. The first allows processing all images in the root folder, while the second only processes the current institution’s answer sheet images. The first button warns the user that the process will require a lot of time since many answer sheet images will be processed, while the second button omits this warning. Finally, the bottom of the central panel shows a read-only grid, displaying the recognition result for each processed answer sheet image, whether the user chose to process the root folder or only the current institution.

## 4. Results

### 4.1. Circle-Detection Comparison with Circle Hough Transform

To obtain a reference for the accuracy achieved by a known method, we first analyzed the dataset to count the number of circles in the answer section of the answer sheet, excluding the circles in the ID section from the analysis. Under ideal conditions, we can assume that only a quarter of the total circles should be marked; however, the dataset does not reflect this trend due to common errors in marking by applicants, including multiple marks and unmarked answers, which complicate the estimation of marked and unmarked circles in the dataset. To estimate this distribution accurately, we analyze manual reviews conducted by our volunteers to calculate the total number of circles in the answer sheets, as well as the distribution of unmarked and marked circles. Table 2 shows the total, marked and unmarked circle distribution in the dataset, and the corresponding accuracies of the CHT and the proposed algorithm.

In this section, we describe tests using the Circle Hough Transform (CHT) detection algorithm on selected answer sheets from the dataset. The CHT algorithm is a standard tool for circle-detection applications, but there are some problems encountered in its execution. To adequately detect the regions of the answer sheet, we first apply adaptive thresholding, which allows us to overcome slight differences in lighting. Later, we use a dilation to thicken the edges of the circles and finally apply the CHT algorithm to find the circular regions. Under ideal conditions (the marks do not exceed the circle boundaries), this algorithm performs adequately. However, suppose the applicant exceeds these marks or does not fill in the marks adequately. In that case, the algorithm fails to locate these marks. Therefore, it is necessary to adapt another algorithm to overcome these failures.

To support the results obtained, Figure 11 shows some cases where the CHT transform fails and where our algorithm succeeds. We briefly describe these cases below.

Where the applicant exceeded the space available to mark the circle, extending their mark outward in a uniform or non-uniform manner. Figure 11a shows that the CHT algorithm fails to detect the mark in responses 3, 5, 6, 7, 8 and 39, while our algorithm successfully detects them (see Figure 11b). The failure of the CHT algorithm is attributable to the applicant not filling in the selected circle for the answer, or in some cases, exceeding the limits of that circle.Where the applicant joined several circles with pencil strokes, causing the CHT algorithm to fail and our algorithm to succeed. Figure 11c,d illustrate a case where both the CHT algorithm and the proposed algorithm yield the same responses. Still, the CHT algorithm fails in response 38, given that the applicant generated a horizontal line connecting the circles in that response. In this particular case, the proposed algorithm successfully performs circle identification, and the output response is unmarked (X), as none of the options are selected.Where the applicant did not fill in the selected circle, making it impossible for the algorithm to determine whether or not the mark was present. The CHT algorithm fails in responses numbered 6, 33, 34, 41, 68, 70 and 71 (see Figure 11e), while our algorithm successfully detects the circles in those same responses (see Figure 11f). The CHT algorithm fails because the applicant mismarked the space for that particular response.

### 4.2. Individual Answer Item Accuracy and Error Analysis

We analyzed the dataset based on the output obtained from the analysis performed by the proposed system and the manual verifications carried out by our volunteers, generating statistics related to the class distribution and the accuracy (and error) for each class. For example, for a test with 90 items (groups of four circles grouped into three columns with 30 groups each), the class of each item can be one of 4 options (A, B, C or D depending of the option marked by the applicant), or X (meaning that the student left that item unanswered) or M (meaning that the student marked two or more options, regardless of which ones). Figure 12 illustrates the distribution of the dataset according to the previously defined classes. It is essential to note that classes X and M are unbalanced because they represent cases that fall outside the ideal scenario and are mutually exclusive (it is not possible to have multiple answers or none at the same time), with the multiple-answer class (M) having the fewest instances in the dataset. Figure 13 presents the confusion matrix obtained from the analysis of the dataset, summarizing the overall and class-specific performance of the proposed system. Finally, Figure 14 shows the accuracy and error of the system for each of the classes mentioned above. The system performs remarkably well (greater than 99.4%) for classes related to a single response or when there is no response from the applicant. However, its worst performance is in cases where there are multiple responses, which are more difficult to detect, but still within an acceptable range (greater than 80%).

### 4.3. Answer Sheet Accuracy and Error Analysis

This section presents the results concerning the algorithm for identifying the responses of the digitized answer sheets processed by the proposed system. The application processed the 6029 images, and we used an accuracy metric to obtain the percentage of exams with all the answers (90 or 100) correctly identified without error. If the algorithm found at least one unidentified answer in the answer sheet, this exam is considered an error. We used the same technique to obtain the percentage of exams without ID errors. Table 3 summarizes the results obtained by the proposed application.

To estimate this accuracy, it was necessary to review the total number of exams and their corresponding answers generated by the proposed system. For this reason, we handed out a set of answer sheets to a group of assistants who reviewed the answers marked by the students and those generated by the system, which helped us identify errors. In this way, it was possible to identify cases in which the algorithm incorrectly recognized the marks. We will acknowledge the work of these volunteers in the acknowledgments.

We analyze the exams the system classifies as not detecting 100% of their answers correctly. Table 4 shows the result of this analysis. Of the 232 exams with at least one error, 166 have exactly one error, 25 have two errors and so on, as reported in Table 4. The maximum number of answer errors in an exam is 17. The distribution of answer errors is loaded to have a low number, specifically those with only one digit, since errors with two digits occur only 1 time (12, 14 and 17 errors). In contrast, the most common answer error is one, representing an error of 0.011 for a 90-answer test or 0.010 for a 100-answer exam.

In this paper, we report an MCQ answer sheet recognition system. We start from a dataset with 6029 images of answer sheet scans obtained from 44 high schools throughout the state of Tamaulipas, having between 90 and 100 answers per test. We counted the total number of answers in the dataset, giving 564,040 answers.

As a prerequisite to analyzing the images, we defined a hierarchy of directories where a root folder must contain the folders of the institutions to be processed, and each institution must have a subfolder for the three grades to be evaluated (tenth, eleventh and twelfth). Each grade folder should contain the images to be analyzed.

We implemented a desktop application that allows the operator to select either the root directory, an institution’s directory or a grade’s directory. Once the directory is analyzed, the system starts the analysis of the images and generates a table (using a CSV format) with the answers obtained from the exam. Before implementing the system, we performed a performance comparison to estimate accuracies that validate the proposed system’s usability.

### 4.4. Hard Cases Considered by the Proposed OMR Algorithm

The following is a list of common problems encountered by the proposed mark recognition algorithm in answer sheets, specifically due to causes unrelated to the image-processing stage. We can group these problems in the answer sheets into the following types: errors caused by the scanning process, which include incorrect orientation, page folding or scanning artifacts; errors caused by the assessed student, who could mismark the answers or include some annotations that invade the regions designated for the recognition of the marks. We describe those cases where it is possible to resolve the problem.

The handwritten text added by the student interferes with the mark-identification algorithms since its location generates false marks in undesired places. Ideally, it would be necessary to detect this text after scanning the test sheet to avoid the interference mentioned above. Figure 15 shows examples of this problem.Another problem detected by the mark-detection algorithm is the folding of the answer sheet, which may have occurred accidentally during the scanning process. The mark recognition discards these answer sheets since the folding causes the algorithms to obtain a different number of marks than the one that should be present. Figure 16 show examples of this problem.In isolated cases, students exceed the limits of the circles when marking the answer sheet, and in some cases, the marking appears blurry. In most cases, the algorithm considers the mark as valid. Figure 17 shows answer sheets with this problem.In some isolated cases, students make an incorrect filling of the answer sheet; instead of filling the circle, they only put a diagonal mark, or the filling of the circle is not complete. In most cases, the algorithm considers this type of mark as valid. Figure 18 shows examples where the student marked the response circle incorrectly or very faintly on the response sheet. Figure 18a shows a fragment of an answer sheet image with some situations the OMR algorithm faced. Although the test applier provided the instruction that students must fill the circles of the answer sheet, they omit this indication and only mark a part of the circle (more precisely, in answers 31 to 38, among others), complicating the algorithm in charge of detecting whether there is a mark in the circle or not. Another indication is that they should not go beyond the margins of the circle, but again this indication is issued (more precisely, in answers 1, 2, 6 and 7, among others).Each high school scans the answer sheets. This scanning can be susceptible to errors, such as scanning the sheets in the wrong orientation or introducing artifacts during digitization, including lines that do not belong to the original answer sheet. Figure 19 shows answer sheets with this problem.During the exam, a student may change the answer to a given question. The erasing process may affect the visibility of adjacent marks, complicating the recognition process and leaving an imprint on the mark with a smudge. Figure 20 shows examples of this problem.

### 4.5. Detected Answer Report

Figure 21 shows an image of a 90-answer sheet (left) and its generated answer report (right). We used the answer report to verify that the OMR algorithm accurately detects the answer marks on each answer sheet in the dataset. In the answer sheet, a gray rectangle encloses the circle options where the assessed student did not select any option. A light gray line underlines the case where the assessed student selected several options for an answer. The right side of the image shows the corresponding output of the OMR algorithm, where the system assigns X (unselected) when the assessed student did not mark any option for a specific answer. In contrast, the system assigns M (Multiple answers) when the assessed student selects several options for the same answer.

### 4.6. Analysis of Alternative Process Usage

Figure 22, Figure 23 and Figure 24 present an analysis of the distribution of alternative process usage about the main algorithm. The main algorithm of the IDASD located the contours of the ID and answer sections in 97.1% of the analyzed answer sheets in a first pass. The remaining percentage required the alternative processes 1, 2 or 3, as shown in Figure 22a. The rotation of the answer sheets depends on the contours obtained by IDASD. Once rotated, in a second pass, we execute the IDASD again to re-detect the contours of the ID and answer sections. In this case, the IDASD detects the ID and answer section in 97.15% of the answer sheets using the main algorithm, while alternative processes 1, 2 or 3 detect the contours of the remaining percentage, as shown in Figure 22b.

The ASP process analyzes the responses marked by the user within the contour detected by the IDASD. The main algorithm of the ASP locates the answer marked by the students in 98.32% of section 0 of analyzed answer sheets. The remaining percentage required one of the alternative processes 1, 2, 3 or 4, as shown in Figure 23a. Analogously, the main algorithm of the ASP locates the answer marked by the students in 98.13% of section 1 of analyzed answer sheets. The remaining percentage required one of the alternative processes 1, 2, 3 or 4, as shown in Figure 23b. Finally, the main algorithm of the ASP locates the answer marked by the students in 98.32% of section 2 of the analyzed answer sheets. The remaining percentage required one of the alternative processes 1, 2, 3 or 4, as shown in Figure 24a.

### 4.7. Execution Time Performance vs. Manual Answer Extraction

Regarding the execution time, we ran the mark-extraction process on the 6029 answer sheets of the dataset, obtaining a processing time of 1 h and 45 min. We achieved an average processing time of 1.05 s, equivalent to processing 57 exams per minute. For a rough comparison, we asked the volunteers to measure the minimum time (in ideal conditions) to obtain the exam items manually, yielding an average of 2 min (120 s). The latter allows us to conclude that our process is 114.28 times faster (or at least two orders of magnitude) than the manual method. We measured the execution time using a PC (Dell technologies, Round rock, TX, USA) equipped with an Intel Core i7-4770 processor at 3.9 GHz and 8 GB of RAM.

## 5. Conclusions and Future Work

In this paper, we developed an application for OMR on scanned MCQ test answer sheets. This application includes alternative processes tailored to specific cases identified in this dataset. The latter could be its main drawback, but the results allow us to conclude that the algorithm covers the most representative cases. Another aspect to consider is that there are atypical cases where the algorithm yields more than 10 errors in the detected answer sheets, as these tests exhibit outlier characteristics not covered in the proposed alternative processes. Of the cases in which the algorithm detected incorrect answers, the most common was when only one of the answers was incorrectly detected, which represents 2.75% of the total number of test sheets, equivalent to 166 of the 6029 answer sheet images in the dataset used in this article. Other cases involving errors in exams of up to two digits include instances of 2 or 3 wrong answers, while in the rest, this error tends to be of one digit or zero. The latter is not statistically relevant to the student’s grade. There are some outlier cases in which, due to situations not considered by the alternative processes, the proposed algorithm yields between 10 and 17 incorrect answers (3 cases in total, representing 0.049% of the total), which could impact the student’s grade. The latter requires a more detailed study to identify the causes of such failures. We observed that most of the above errors are caused by the assessed student not following the instructions to select the options correctly.

The proposed algorithm is a robust OMR system that has been extensively tested on real-world answer sheets, achieving over 99% accuracy on a dataset of more than 6000 images without requiring a training stage for exams with the same layout. Unlike complex approaches such as convolutional neural networks, it is based on well-established image-processing techniques, allowing for a clear understanding of each processing stage, easier error detection and lower computational demands. Its design avoids the need for high-performance hardware, making it practical for use on standard office computers, while still offering efficient performance with an average processing time of about one second per answer sheet.

We tested the algorithm proposed in this paper with 6029 answer sheet images, measuring performance by two metrics. The first is the number of exams where 100% of the answers were successfully detected, obtaining 96.15% of the total. The second metric we considered involves the percentage of the total number of options contained in the 6029 exams, totaling 564,040 options, of which the algorithm detected the correct answer in 99.95% of the cases.

To measure the processing time of the exams, we ran the algorithm for the 6029 exams, obtaining a time of 1.05 s per exam, and compared it against the time it would take a human operator to obtain the selected answers, estimated at 2 min, representing that our process is 114.28 faster compared to the human operator. The obtained execution time would enable a large number of tests to be evaluated in a short time, yielding results with minimal human intervention.

The system demonstrates the capacity to handle a significant variability in input information, particularly concerning the devices used for scanning the answer sheets and the errors introduced by assessed students when selecting test options. The proposed system is flexible concerning the layout and orientation of the answer circle distribution on the sheet. If necessary, to make adjustments, such as adding columns, deleting items or extending the ID section, it is possible to modify the system parameters by making minimal adjustments.

One potential improvement is to integrate the current system that extracts test answers with the existing system used to generate MCQ tests, thereby obtaining students’ grades. Another improvement is to develop a mobile application that analyzes the answer sheet at the time the student finishes the test to issue an alert when the answer sheet presents some of the issues in the complex cases previously described, where the student does not correctly mark the answers, even comment that the student has unanswered answers, or even verify that the ID obtained from the answer sheet coincides with the ID assigned to the student.

We did not design the proposed algorithm for real-time recognition of marks on answer sheets, as this would require significant computational resources, even for the adapted algorithms. To achieve real-time performance, it would be necessary to optimize the algorithms used; however, this would also involve restricting the capture conditions and adding a high-definition digital single-lens reflex (DSLR) camera to obtain the input answer sheet images. The runtime of our algorithm is sufficiently low to be used in a processing chain that includes scanning and detecting marks on answer sheets on a massive scale, as it would only add a fraction of the time currently required to digitize the images.

In the preprocessing stage, it is possible to incorporate state-of-the-art algorithms that can reduce the noise caused by scanning answer sheets under a wide variety of lighting conditions. According to the analysis of the response sheet images of our dataset, we did not observe a need to add global denoising techniques. However, we could evaluate the effect of using advanced noise removing techniques, such as the Variational Nighttime Dehazing framework using Hybrid Regularization VNDHR [30], designed to improve visibility in hazy nighttime scenes, but with applications to remove noise from low-light conditions while maintaining sharp object boundaries. Another noise-detection algorithm is Global Image Denoising (GID) [31], which exploits the non-local self-similarity property of images by searching the entire image to find similar patches, grouping them and then applying filtering to reduce noise while preserving structure.

In the recognition stage, one improvement we can incorporate into the system is the use of convolutional neural networks to perform a more accurate classification of responses, specifically by including YOLOv8 as a detector and segmenter of marks. Using CNNs is feasible, but it requires a considerable amount of data preparation, including segmenting individual items and matching them with the manual labeling available in the dataset. It is also possible to train a specialized CNN model using the obtained dataset to improve the current accuracy of the proposed system.

## Figures and Tables

**Figure 1 jimaging-11-00308-f001:**
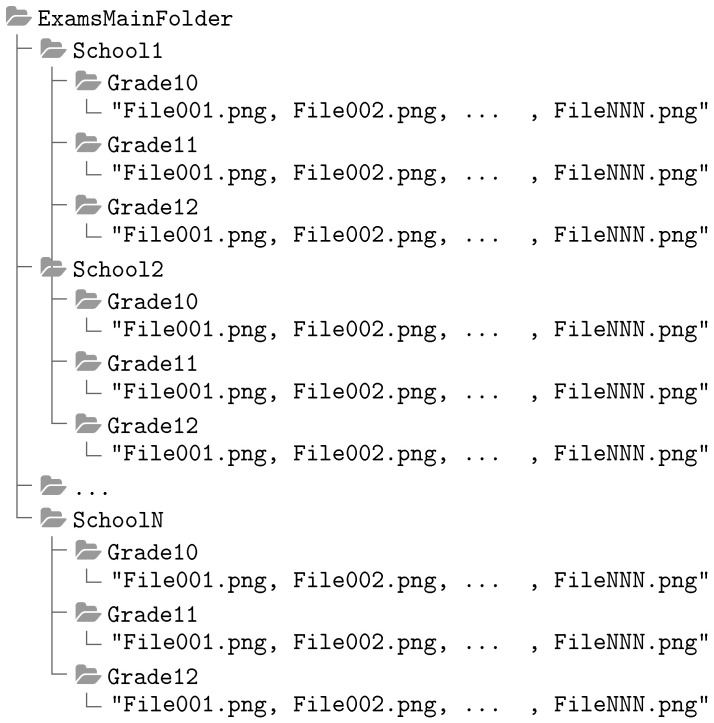
Folder organization tree for the automatic grading app.

**Figure 2 jimaging-11-00308-f002:**
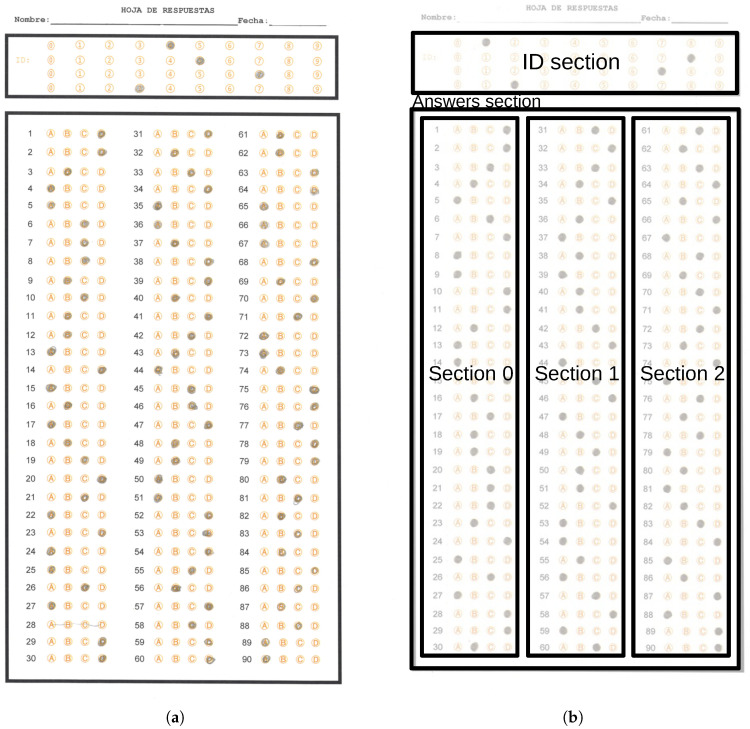
Examples of an answer sheet and section distribution. (**a**) Answer sheet example of the dataset used in this work. (**b**) Section distribution presented over an answer sheet example.

**Figure 3 jimaging-11-00308-f003:**
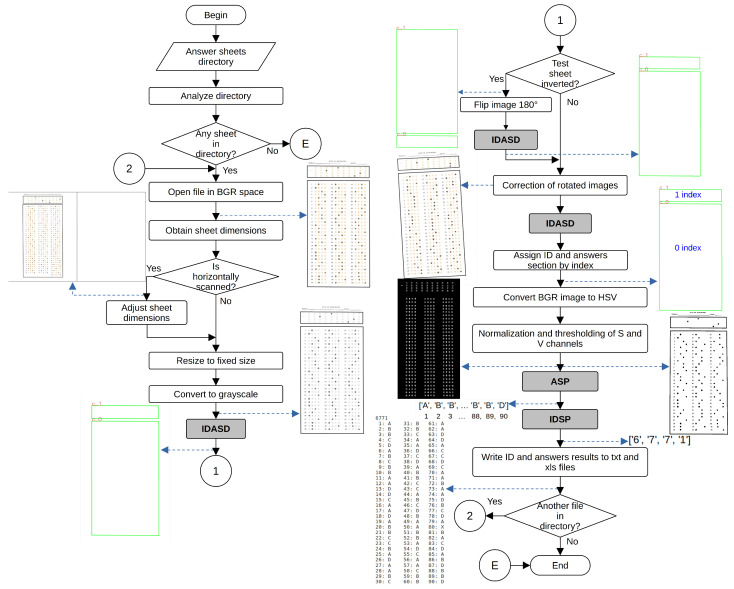
Main dataflow of the proposed system.

**Figure 4 jimaging-11-00308-f004:**
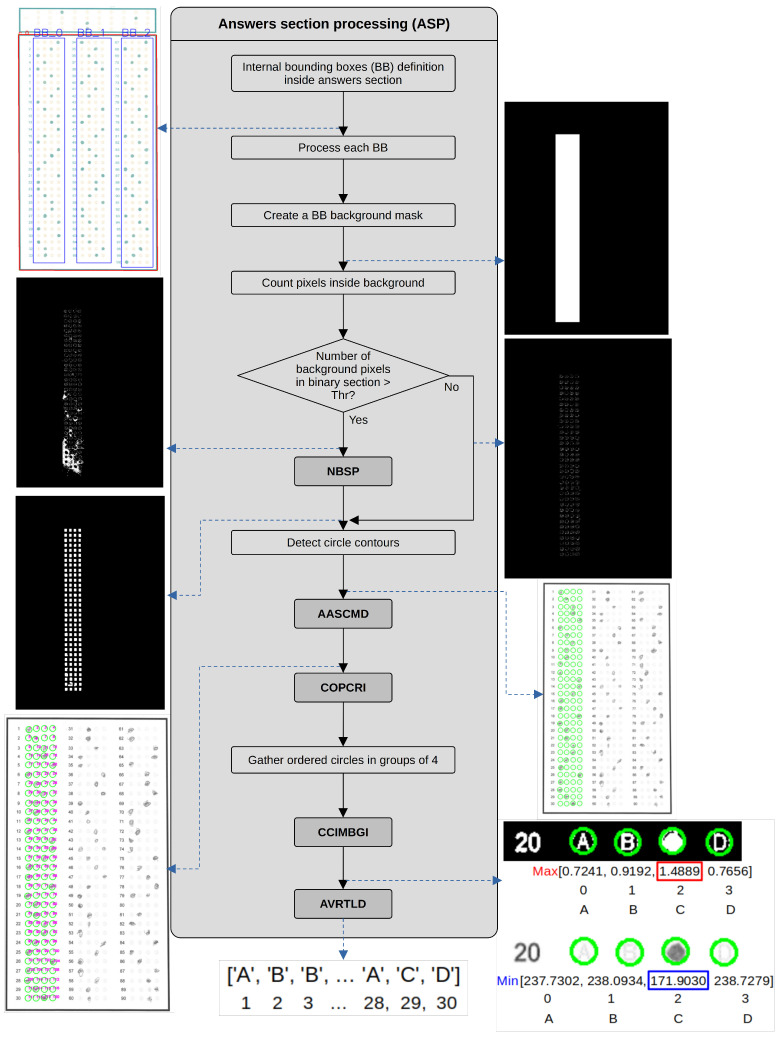
Dataflow of the ASP process.

**Figure 5 jimaging-11-00308-f005:**
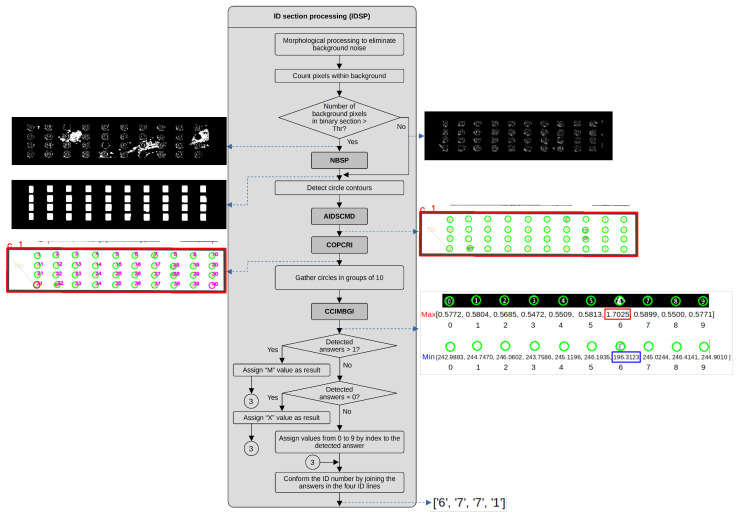
Dataflow of the IDSP process.

**Figure 6 jimaging-11-00308-f006:**
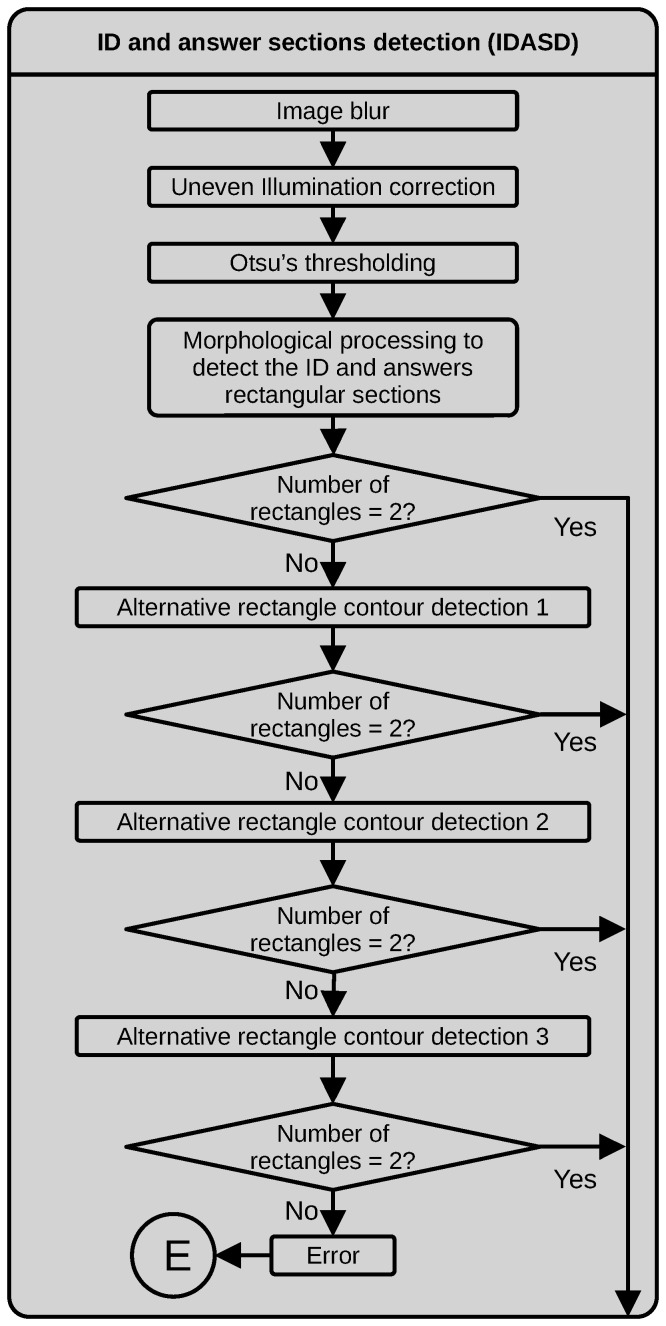
Dataflow of the IDASD.

**Figure 7 jimaging-11-00308-f007:**
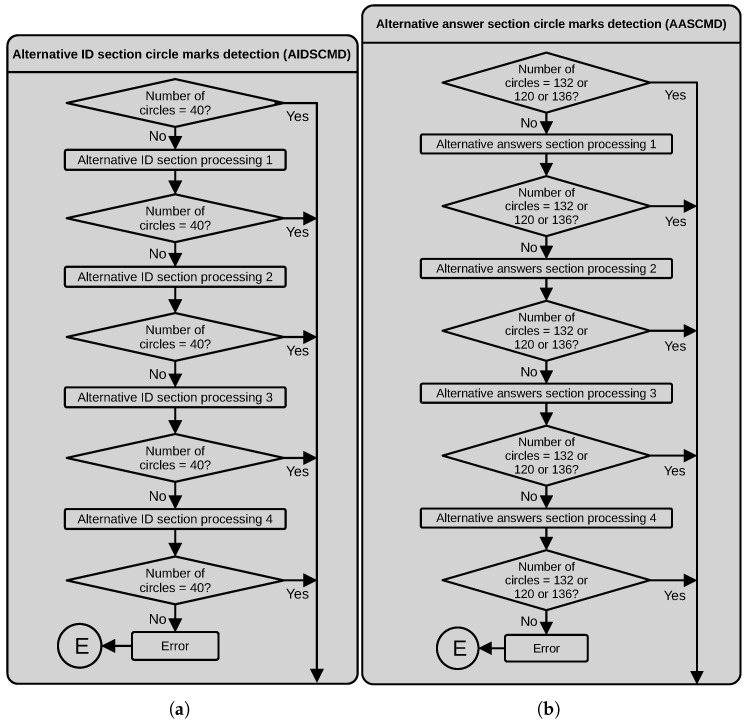
Dataflow of the (**a**) AIDSCMD and (**b**) AASCMD processes.

**Figure 8 jimaging-11-00308-f008:**
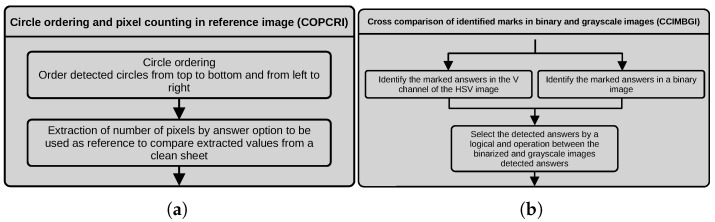
Dataflow of the (**a**) COPCRI and (**b**) CCIMBGI processes.

**Figure 9 jimaging-11-00308-f009:**
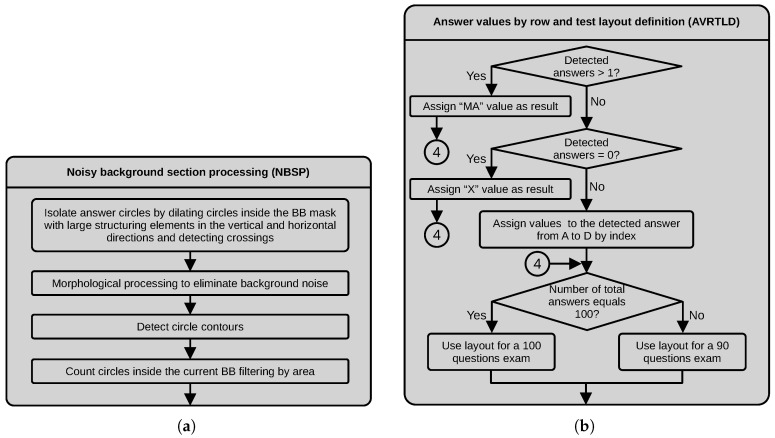
Dataflow of the (**a**) NBSP and (**b**) AVRTLD processes.

**Figure 10 jimaging-11-00308-f010:**
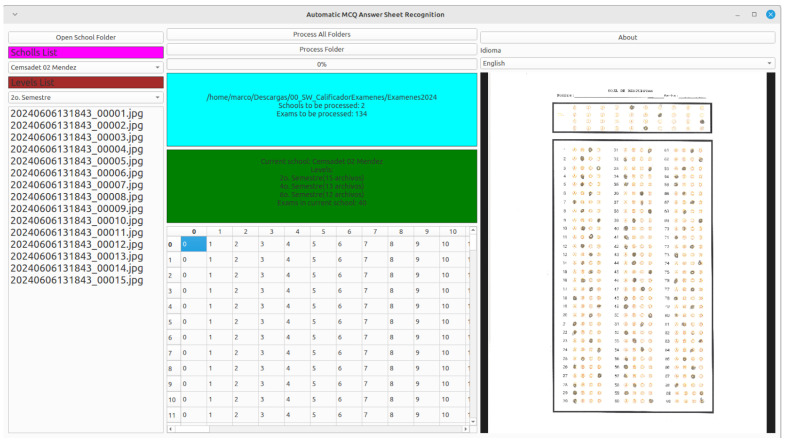
Automatic MCQ answer sheet recognition. On the left side is shown the list of the answer sheets, including the list of schools and grades. In the center part, the cyan colored panel shows the selected folder, including the contained schools and the number of total exams, and the green colored panel shows details of the chosen school, specifically the distributions of exams with respect to grades. At the bottom is the result of the response collection in tabular format. The right side shows a view of the answer sheet, which changes as the user selects a different test from the list on the right side.

**Figure 11 jimaging-11-00308-f011:**
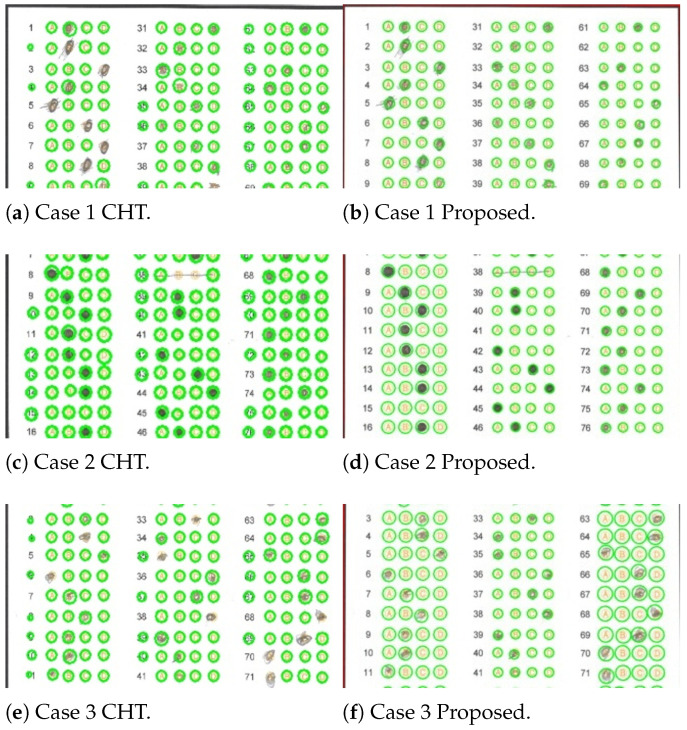
Selected answer sheets used for a comparison of the proposed algorithm (**b**,**d**,**f**) vs. Hough detection (**a**,**c**,**e**).

**Figure 12 jimaging-11-00308-f012:**
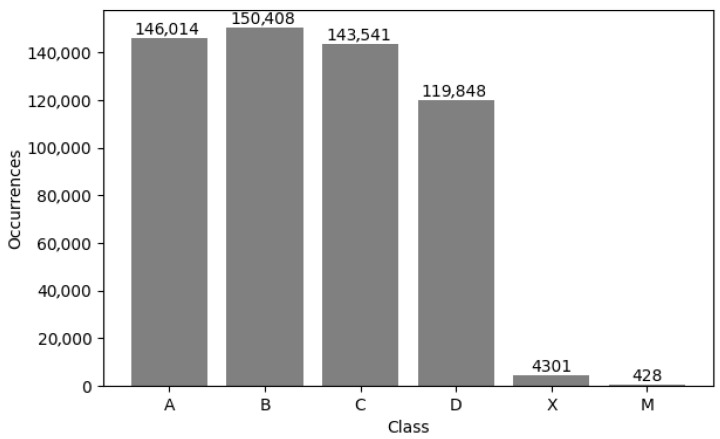
Class occurrence distribution.

**Figure 13 jimaging-11-00308-f013:**
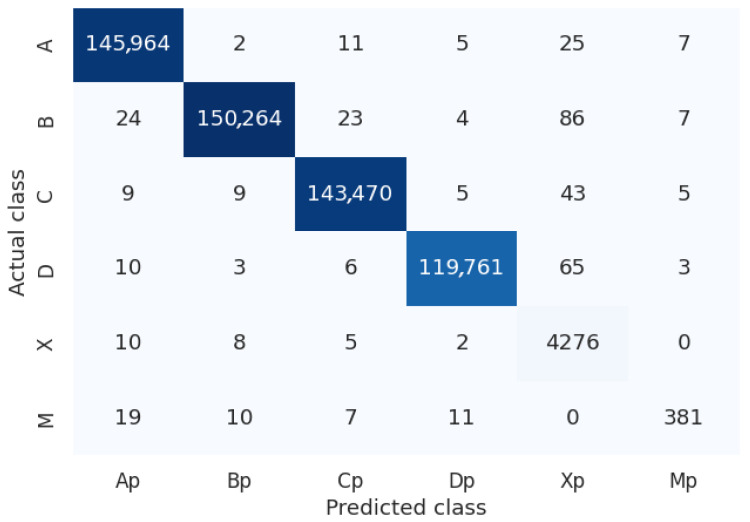
Confusion matrix. The dark colors on the diagonal represent the number of correct predictions made by the proposed system for the considered class. The darker the blue color, the higher the number of correct predictions made by the proposed system.

**Figure 14 jimaging-11-00308-f014:**
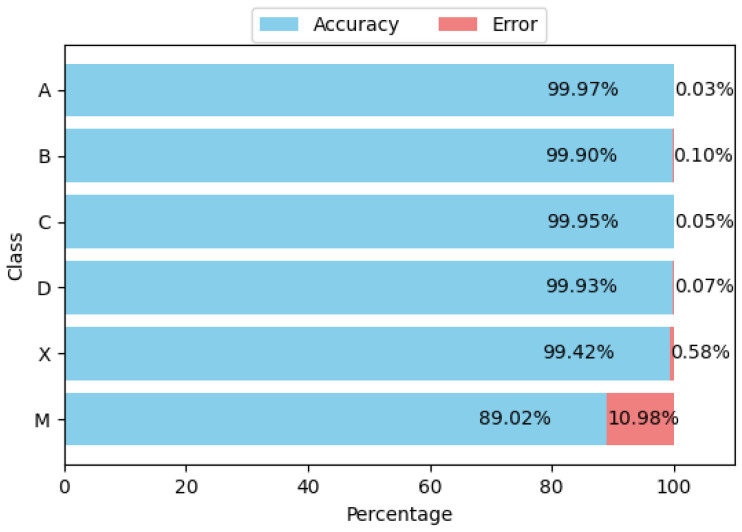
Accuracy and error per class.

**Figure 15 jimaging-11-00308-f015:**
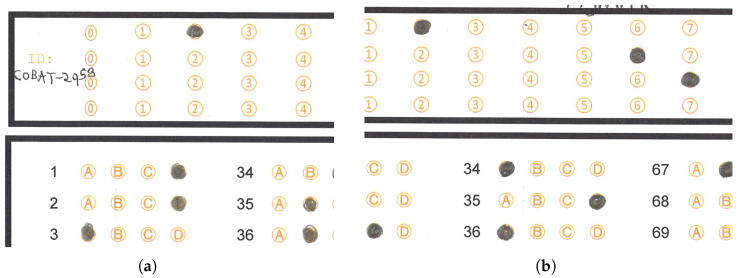
Examples of handwritten text annotations in processed answer sheets. (**a**) Handwritten text over a circle option. (**b**) Handwritten text over the surrounding rectangle of the ID section.

**Figure 16 jimaging-11-00308-f016:**
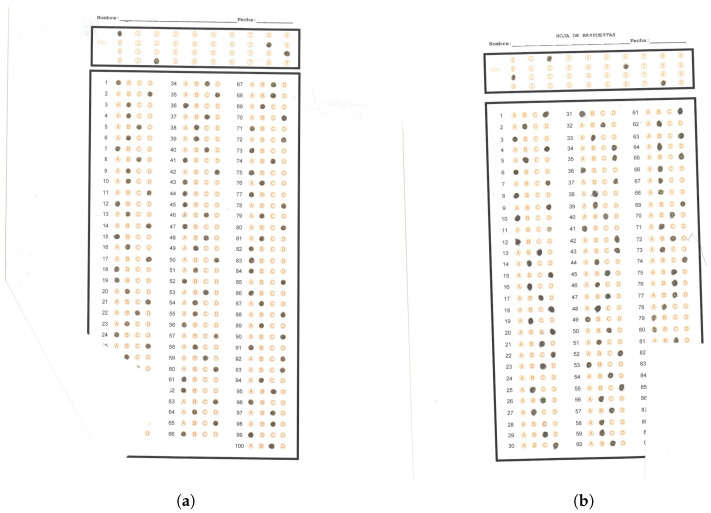
Examples of folding in answer sheets caused by a wrong scanning process. (**a**) Folded bottom left corner in a scanned answer sheet. (**b**) Folded bottom right corner in a scanned answer sheet.

**Figure 17 jimaging-11-00308-f017:**
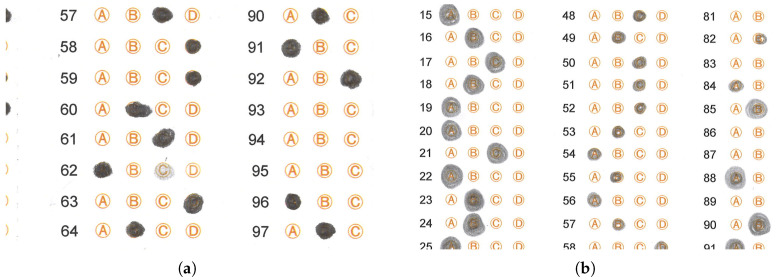
Answer sheets with marks exceeding the space of the circle provided for the filling in of such an answer or different size marks. (**a**) Answer sheet with marks outside the option circles and different tonalities. (**b**) Different answer marks sizes in the same answer sheet.

**Figure 18 jimaging-11-00308-f018:**
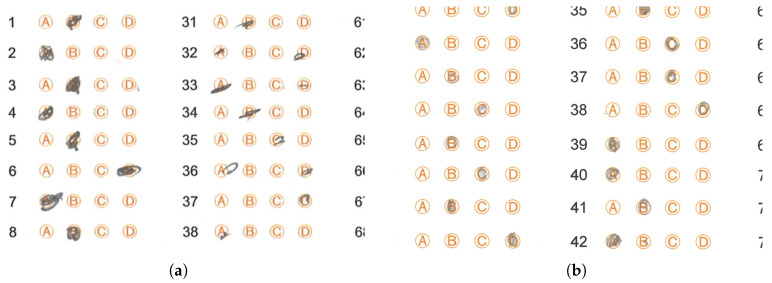
Answer sheets with incorrectly or very faintly marked circle options. (**a**) Incompletely filled circle options. (**b**) Circle options filled with faint marks.

**Figure 19 jimaging-11-00308-f019:**
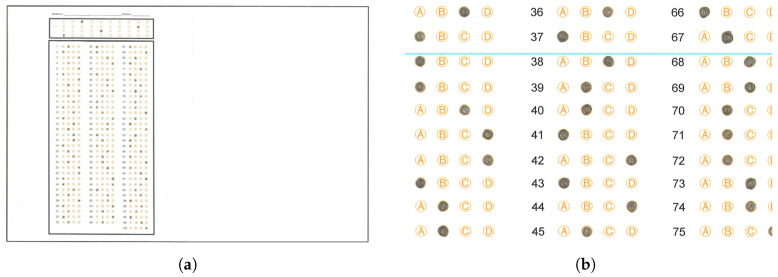
Rotated answer sheet or with scanning artifacts. (**a**) Image scanned in landscape orientation, so it only appears in the first half of the scan space. (**b**) Blue artifact over the circle option answers.

**Figure 20 jimaging-11-00308-f020:**
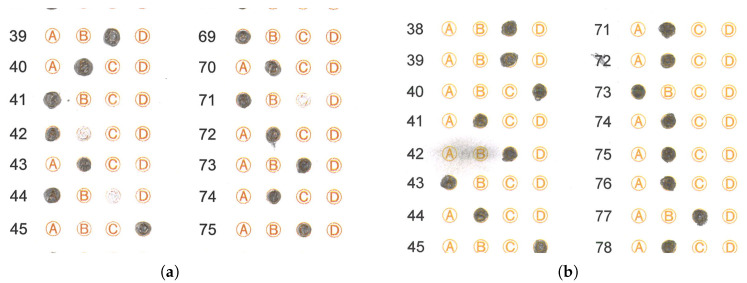
Answer sheets with erased circle options or dark smudges that could affect the mark recognition process. (**a**) Erased circle options. (**b**) Dark smudge over circle options.

**Figure 21 jimaging-11-00308-f021:**
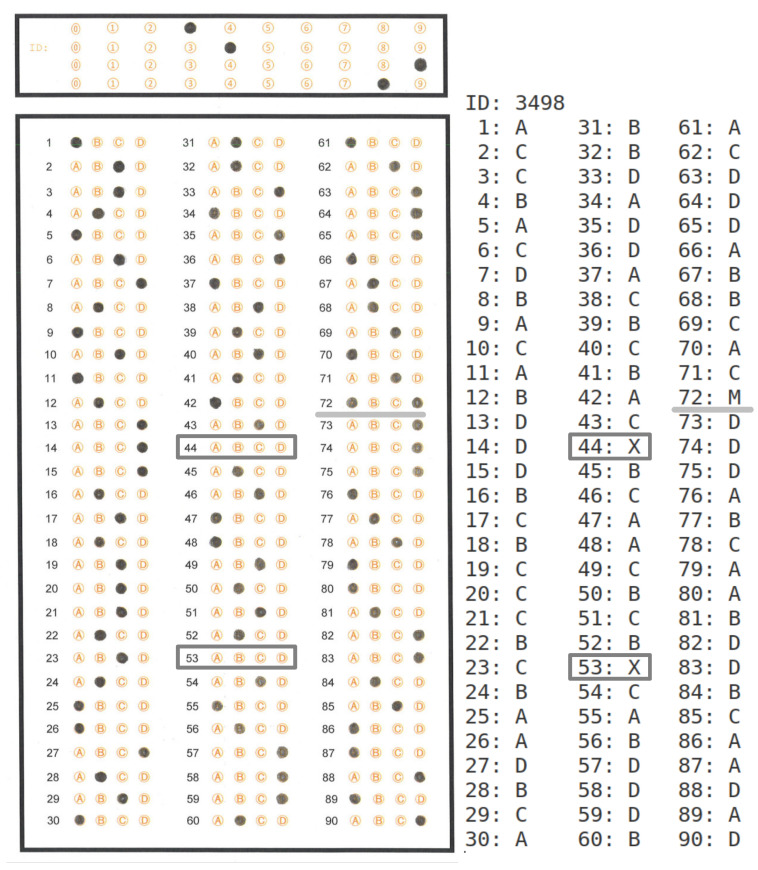
A 90-answer sheet image and its corresponding answer report. The dark gray-framed answers are examples of unanswered items. The light gray underlined item is an example of a multiple-marked item.

**Figure 22 jimaging-11-00308-f022:**
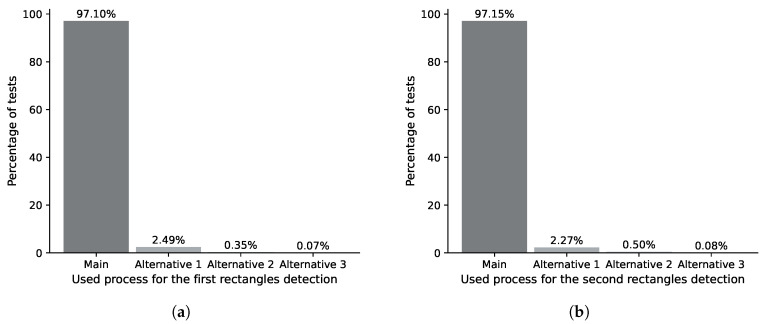
Percentage distribution of the use of algorithms for detecting ID and response section contours. (**a**) Percentage distribution for the first pass of the ID and answer section contours. (**b**) Percentage distribution for the second pass of the ID and answer section contours.

**Figure 23 jimaging-11-00308-f023:**
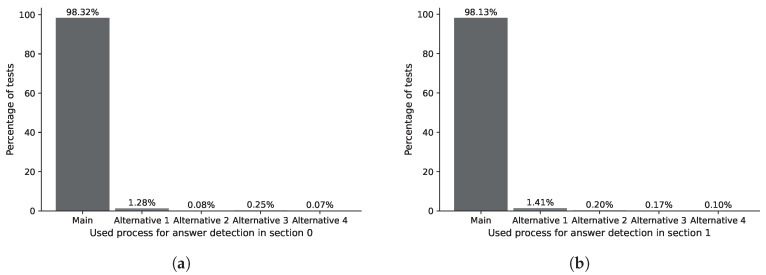
Percentage distribution of the use of algorithms for locating answer marks in the answer section. (**a**) Percentage distribution for locating answer marks in section 0. (**b**) Percentage distribution for locating answer marks in section 1.

**Figure 24 jimaging-11-00308-f024:**
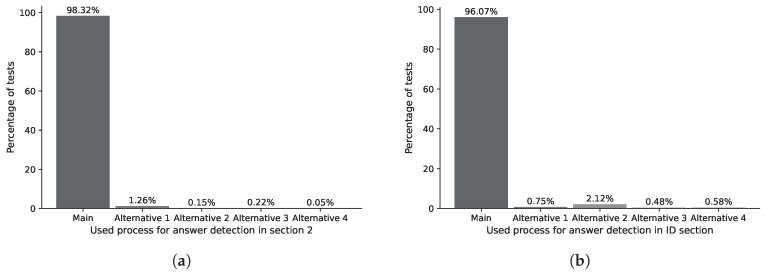
(**a**) Percentage distribution for locating answer marks in section 2. (**b**) Percentage distribution for locating answer marks in ID section.

**Table 1 jimaging-11-00308-t001:** Dataset statistics.

Grade	Number of Tests
10th grade	2157
11th grade	2143
12th grade	1729
Total	6029
Answers distribution over the dataset	
90 answer-tests	3886
100 answer-tests	2143
Total institutions	44
Total answers	564,040

**Table 2 jimaging-11-00308-t002:** Dataset circle distribution and detection accuracies.

Circle Type	Total Circles	Recognized Circles and Accuracy	
		CHT	Proposed
Marked circles	546,978	171,094 (31.28%)	533,030 (97.45%)
Unmarked circles	1,709,212	1,648,364 (96.44%)	1,707,160 (99.88%)
Total circles	2,256,160	1,819,458 (80.64%)	2,224,190 (99.26%)

**Table 3 jimaging-11-00308-t003:** Accuracy of the proposed system.

Error in answers section	
Exams without error	5797
Exams with at least one error	232
Accuracy	96.15%
Error in ID section	
Exams without error	6000
Exams with ID error	29
Accuracy	99.50%
Error in answers compared to the entire dataset	
Total answers	564,040
Answer detected with error	424
Accuracy	99.95%

**Table 4 jimaging-11-00308-t004:** Answer error distribution.

Summary of Answer Errors	
Exams with 1 answer error	166
Exams with 2 answer errors	25
Exams with 3 answer errors	14
Exams with 4 answer errors	9
Exams with 5 answer errors	8
Exams with 6 answer errors	4
Exams with 7 answer errors	1
Exams with 8 answer errors	2
Exams with 9 answer errors	0
Exams with 10 answer errors	0
Exams with 11 answer errors	0
Exams with 12 answer errors	1
Exams with 13 answer errors	0
Exams with 14 answer errors	1
Exams with 15 answer errors	0
Exams with 16 answer errors	0
Exams with 17 answer errors	1

## Data Availability

The original contributions presented in this study are included in the article. Further inquiries can be directed to the corresponding author.

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
