# Peer review of "Unsupervised Optical Mark Recognition on Answer Sheets for Massive Printed Multiple-Choice Tests"

_2313-433X, 2025, doi:10.3390/jimaging11090308_

Round 1

Reviewer 1 Report

Comments and Suggestions for Authors

In this paper, the authors present a computer program that was used to solve the task of  extracting answers from the answer sheets. In practice, a paper is rather a manual or the technical report for this desktop application. The authors demonstrate how data files are stored in directories (Section 3.1), basic directory operations (Section 3.2), and a very basic image-processing algorithm that utilizes the HSV color model, thresholding, and image morphology. What is more, it seems that the algorithm has hardcoded values and rules that are tuned to operate only on this particular dataset (see Section 3.2.1 and following). For this reason, the solution presented in this paper cannot be directly applied to solve any other problem. I appreciate the authors' work on solving this engineering task, but the article they sent is not a scientific paper that can be published in a high-impact scientific journal such as the Journal of Imaging.

Author Response

Comment 1:

Reviews: 

The main contribution of our work is that we found the combination and parameterization of well-known image processing algorithms to solve the mark recognition problem, which are lightweight enough to be used on low-performance computers. We proposed an approach that works with scanned answer sheets from heterogeneous sources and achieves a high mark recognition rate compared to other algorithms. We have conducted additional tests, including a comparison of the Circle Hough detection algorithm for detecting circles in the answer items (Section 4.1, not present in the original manuscript version). In addition, we estimated the accuracy of mark recognition for the collected dataset, obtained the accuracy and error rates per class, and analyzed the distribution of errors and hits in the form of a confusion matrix (Section 4.2, not present in the original manuscript version). The methodology does not require a training set of images, unlike more complex algorithms based on convolutional neural networks. The disadvantage of our approach is that if the layout of the answer sheet changes, we must adjust the parameters to fit this new layout. We have added a bulleted list highlighting the main contributions of the article at the end of the introduction section.          

While it is true that we experimentally estimate the intensity and size thresholds of the morphological kernels, these thresholds depend on the dimensions of the bounding boxes and the average vertical and horizontal distances between the circles of the options (items) in the answer sheet format. Because of this, all exams are resized to a single size, ensuring that these thresholds are consistent for the whole dataset. To determine the threshold selection that determines the presence or absence of noise within the bounding boxes, we analyzed the number of pixels in the background of a set of scanned sheets without response marks using near-ideal illumination conditions. We averaged the number of pixels for these clean images. If the answer sheet layout changes, we need to recalculate these thresholds; however, the overall algorithm would not undergo any change. We added a paragraph in subsection 3.2.6 to explain the latter. 

We have made modifications to the article to compare with existing image processing techniques. Regarding circle detection, we have included a comparison of the proposed algorithm with the Circle Hough Transformation, which is considered a standard for detecting circles. We analyze cases in which CHT fails, but our algorithm successfully detects circles, particularly those that have been marked, thereby validating our proposal. We generate dataset-level statistics regarding this comparison, highlighting that our algorithm is successful in cases where CHT fails, specifically in instances where marks are not uniform. 

Due to time constraints, we were unable to compare our method with one based on convolutional neural networks. However, we have established in our future work that a potential improvement would be to use the generated dataset to train a model based on convolutional neural networks that is much less sensitive to noise and also achieves greater accuracy, both in circle detection and mark detection.

Reviewer 2 Report

Comments and Suggestions for Authors

This paper proposes desktop application for optical mark recognition (OMR) on the scanned multiple-choice question (MCQ) test answer sheets. Although the proposed method has significant practical value, several major concerns should be addressed.

  1. I understand that the authors employ the unsupervised OMR for answer sheets. However, the main contributions should be summarized as several points in a bullet-point fashion. Moreover, highlighting the differences between the proposed method and existing techniques should he provided.
  2. In Figure 3-4, the authors use the text to show the workflow of the proposed method. Could the authors use the images (visualization) instead of the text to show the flowchart of the proposed method?
  3. If the captured images are affected by surrounding imaging conditions, leading to degradation such as noise or blurriness for answer sheets, how does your method perform under these circumstances? Therefore, some recent image enhancement, restoration, and denoising methods (1-2), such as VNDHR, UVRM, etc, can be reviewed as the preprocessing techniques to handle these degraded images.

(1) VNDHR: Variational Single Nighttime Image Dehazing for Enhancing Visibility in Intelligent Transportation Systems via Hybrid Regularization

(2) Global image denoising

  1. To my knowledge, several methods can achieve this task for printed multiple choice tests. What are the advantages of the proposed method? Could the authors provide the comparisons between the proposed method and existing methods? It would be better if quantitative metrics could be used to compare and highlight the superiority of your proposed method.
  2. The paper appears more like an experimental report that implements a system for processing answer sheets for massive printed multiple-choice tests, but it lacks academic depth and novelty. For a scholarly publication, the authors should place greater emphasis on the academic contributions and highlight the methodological innovations.
  3. What is the computational time of the proposed method? Is it capable of real-time processing? Under what conditions does the proposed method fail? It would be helpful if the authors could provide failure cases and discuss potential solutions or mitigation strategies.

I recommend major revisions for this paper.

Author Response

Comment 1: "I understand that the authors employ the unsupervised OMR for answer sheets. However, the main contributions should be summarized as several points in a bullet-point fashion. Moreover, highlighting the differences between the proposed method and existing techniques should he provided."

Review: We have added a bulleted list highlighting the main contributions of the article at the end of the introduction section.

Comment 2: "In Figure 3-4, the authors use the text to show the workflow of the proposed method. Could the authors use the images (visualization) instead of the text to show the flowchart of the proposed method?"

Review: We modified Figures 3 and 4 to include images that help readers understand how the proposed algorithms work.

Comment 3: "If the captured images are affected by surrounding imaging conditions, leading to degradation such as noise or blurriness for answer sheets, how does your method perform under these circumstances? Therefore, some recent image enhancement, restoration, and denoising methods (1-2), such as VNDHR, UVRM, etc, can be reviewed as the preprocessing techniques to handle these degraded images. (1) VNDHR: Variational Single Nighttime Image Dehazing for Enhancing Visibility in Intelligent Transportation Systems via Hybrid Regularization (2) Global image denoising"

Review: Given that the system operates with scanned images, blurriness is an artifact created by the applicant, not by lighting conditions. Therefore, we assumed that the scanned answer sheets were free from blur. If it is present, we must incorporate techniques that reduce its effects with minimal impact on mark recognition. According to the analysis of the answer sheet images of our dataset, we did not observe a need to add global denoising techniques. However, we could evaluate the effect of using advanced noise removal techniques, such as those suggested by the reviewer. We mentioned this point in the conclusion and future work section.

Comment 4: "To my knowledge, several methods can achieve this task for printed multiple choice tests. What are the advantages of the proposed method? Could the authors provide the comparisons between the proposed method and existing methods? It would be better if quantitative metrics could be used to compare and highlight the superiority of your proposed method."

Review: We have made modifications to the article. Regarding circle detection, we have included a comparison of the proposed algorithm with the Circle Hough Transformation, which is considered a standard for detecting circles. We analyze cases in which CHT fails, but our algorithm successfully detects circles, particularly those that have been marked, thereby validating our proposal. We generate dataset-level statistics regarding this comparison, highlighting that our algorithm is successful in cases where CHT fails, specifically in instances where marks are not uniform.

Comment 5: "The paper appears more like an experimental report that implements a system for processing answer sheets for massive printed multiple-choice tests, but it lacks academic depth and novelty. For a scholarly publication, the authors should place greater emphasis on the academic contributions and highlight the methodological innovations."

Reviews: We found the combination and parameterization of well-known image processing algorithms to solve the mark recognition problem, which are lightweight enough to be used on low-performance computers. We proposed an approach that works with scanned answer sheets from heterogeneous sources and achieves a high percentage of mark recognition compared to other algorithms. We have conducted additional tests, including a comparison of the Circle Hough detection algorithm for detecting circles in the answer items (Section 4.1, not present in the original manuscript version). In addition, we estimated the accuracy of mark recognition for the collected dataset, obtained the accuracy and error rates per class, and analyzed the distribution of errors and hits in the form of a confusion matrix (Section 4.2, not present in the original manuscript version). The methodology does not require a training set of images, unlike more complex algorithms based on convolutional neural networks. The disadvantage of our approach is that if the layout of the answer sheet changes, we must adjust the parameters to fit this new layout.

Comment 6:  "What is the computational time of the proposed method? Is it capable of real-time processing? Under what conditions does the proposed method fail? It would be helpful if the authors could provide failure cases and discuss potential solutions or mitigation strategies."

Review:  We did not design the proposed algorithm for real-time recognition of marks on answer sheets, as this would require significant computational resources, even for the adapted algorithms. To achieve real-time performance, it would be necessary to optimize the algorithms used; however, this would also involve restricting the capture conditions and adding a high-definition digital single-lens reflex (DSLR) camera to obtain the input answer sheet images. The runtime of our algorithm is sufficiently low to be used in a processing chain that includes scanning and detecting marks on answer sheets on a massive scale, as it would only add a fraction of the time currently required to digitize the images.

Reviewer 3 Report

Comments and Suggestions for Authors

This paper presents a large-scale automated evaluation of multiple-choice questions using traditional image processing techniques under controlled scanning conditions. The system is based on a dataset consisting of 6,029 scanned answer sheets collected from 44 institutions in Tamaulipas, Mexico, containing a total of 564,040 answer instances. The proposed method uses a set of heuristics and alternative fallback procedures to perform preprocessing, ID and answer part detection, perspective correction, and adaptive landmark recognition. The system achieves an accuracy of 99.95% at the level of individual answers and 96.15% at the level of the entire answer sheet. However, there are the following issues that need to be revised.

1.Although the paper surveys a broad range of existing OMR systems, the experimental section does not provide quantitative comparisons with representative baseline methods, such as Hough-based detectors, CNN-based classifiers, or recent object detection models (e.g., YOLOv8). Without such comparisons, the relative advantage of the proposed system in terms of accuracy, speed, and robustness remains unclear. It is strongly recommended to include benchmark experiments on the same dataset against at least one open-source or previously published method, using unified metrics such as precision, recall, F1-score, and execution time.

2.The proposed algorithm heavily depends on manually set thresholds and heuristics (e.g., fixed image resizing, 15% noise pixel threshold, fixed circle counts of 120/132/136). However, there is no analysis of how sensitive these values are to variations in image quality, illumination, or noise levels. A robustness or ablation study should be added to show how parameter tuning affects performance and to identify the critical thresholds that impact accuracy most significantly.

3.The evaluation only reports exam-level and global answer-level accuracy. However, there is no classification or quantification of common error types (e.g., false positives, false negatives, multiple markings, missing marks). Incorporating a confusion matrix or class-wise error analysis would help readers understand which types of misdetections occur and how frequently. This level of detail is essential to assess the reliability of the system in operational environments.

Author Response

Comment 1. "Although the paper surveys a broad range of existing OMR systems, the experimental section does not provide quantitative comparisons with representative baseline methods, such as Hough-based detectors, CNN-based classifiers, or recent object detection models (e.g., YOLOv8). Without such comparisons, the relative advantage of the proposed system in terms of accuracy, speed, and robustness remains unclear. It is strongly recommended to include benchmark experiments on the same dataset against at least one open-source or previously published method, using unified metrics such as precision, recall, F1-score, and execution time."

Review:   Due to hardware constraints in the computer used by the final user for testing the proposed system, we did not consider using a convolutional neural network-based OMR. However, we have considered an alternative version of our system using a CNN-based model trained on the items contained in the dataset and the responses generated by the current system, which could be less sensitive to noise and also achieves greater accuracy in both circle detection and mark detection. We have made modifications to the article. Regarding circle detection, we have included a comparison of the proposed algorithm with the Circle Hough Transformation, which is considered a standard for detecting circles. We analyze cases in which CHT fails, but our algorithm successfully detects circles, particularly those that have been marked, thereby validating our proposal. We generate dataset-level statistics regarding this comparison, highlighting that our algorithm is successful in cases where CHT fails, specifically in instances where marks are not uniform.

Comment 2. "The proposed algorithm heavily depends on manually set thresholds and heuristics (e.g., fixed image resizing, 15% noise pixel threshold, fixed circle counts of 120/132/136). However, there is no analysis of how sensitive these values are to variations in image quality, illumination, or noise levels. A robustness or ablation study should be added to show how parameter tuning affects performance and to identify the critical thresholds that impact accuracy most significantly."

Review: While it is true that we experimentally estimate the intensity and size thresholds of the morphological kernels, these thresholds depend on the dimensions of the bounding boxes and the average vertical and horizontal distances between the circles of the options (items) in the answer sheet format. To determine the threshold selection that determines the presence or absence of noise within the bounding boxes, we analyzed the number of pixels in the background of a set of scanned sheets without response marks using near-ideal illumination conditions. We averaged the number of pixels for these clean images. Because of this, all exams are resized to a single size, ensuring that these thresholds are consistent for the whole dataset. If the answer sheet layout changes, we need to recalculate these thresholds; however, the overall algorithm would not undergo any change. We added a paragraph in subsection 3.2.6 to explain the latter.

Comment 3. The evaluation only reports exam-level and global answer-level accuracy. However, there is no classification or quantification of common error types (e.g., false positives, false negatives, multiple markings, missing marks). Incorporating a confusion matrix or class-wise error analysis would help readers understand which types of misdetections occur and how frequently. This level of detail is essential to assess the reliability of the system in operational environment

Review: We found the combination and parameterization of well-known image processing algorithms to solve the mark recognition problem, which are lightweight enough to be used on low-performance computers. We proposed an approach that works with scanned answer sheets from heterogeneous sources and achieves a high percentage of mark recognition compared to other algorithms. We have conducted additional tests, including a comparison of the Circle Hough detection algorithm for detecting circles in the answer items (Section 4.1, not present in the original manuscript version). In addition, we estimated the accuracy of mark recognition for the collected dataset, obtained the accuracy and error rates per class, and analyzed the distribution of errors and hits in the form of a confusion matrix (Section 4.2, not present in the original manuscript version). The methodology does not require a training set of images, unlike more complex algorithms based on convolutional neural networks. The disadvantage of our approach is that if the layout of the answer sheet changes, we must adjust the parameters to fit this new layout.

Round 2

Reviewer 1 Report

Comments and Suggestions for Authors

The authors addressed all my remarks. In my opinion, paper can be accepted.

Reviewer 2 Report

Comments and Suggestions for Authors

After reviewing the response letter and the revised manuscript, I recommend accepting the paper.

Reviewer 3 Report

Comments and Suggestions for Authors

Thank you for your response. I accept the revised paper.